# Diet in the Pathogenesis and Management of Ulcerative Colitis; A Review of Randomized Controlled Dietary Interventions

**DOI:** 10.3390/nu11071498

**Published:** 2019-06-30

**Authors:** Ammar Hassanzadeh Keshteli, Karen L. Madsen, Levinus A. Dieleman

**Affiliations:** 1Department of Medicine, University of Alberta, Edmonton, AB T6G 2P5, Canada; 2Centre of Excellence for Gastrointestinal Inflammation and Immunity Research (CEGIIR), Edmonton, AB T6G 2X8, Canada

**Keywords:** diet, inflammatory bowel disease, ulcerative colitis

## Abstract

Epidemiological and experimental studies have suggested that diet is one of the environmental factors that contributes to the onset and pathophysiology of ulcerative colitis. Although many patients suffering from ulcerative colitis attribute their symptoms or disease relapse to dietary factors, only a few well-designed randomized controlled trials have been done to investigate the role of diet in the management of ulcerative colitis. Here, we review the potential mechanisms of the relationship between diet and pathogenesis of ulcerative colitis and summarize randomized controlled dietary interventions that have been conducted in ulcerative colitis patients.

## 1. Introduction

Ulcerative colitis (UC)—a subtype of inflammatory bowel disease (IBD)—is a chronic, idiopathic inflammatory disease that affects the colon and is characterized by relapsing and remitting mucosal inflammation [1]. UC patients mostly present blood in the stool and diarrhea [1]. UC is associated with major morbidity in Western countries, and its incidence is increasing in developing countries [2]. The multifactorial pathophysiology of UC includes genetic predisposition, epithelial barrier defects, dysregulated immune responses, microbial dysbiosis, and environmental factors [1,2].

It has been suggested that environmental factors play a major role in the pathogenesis of IBD. Early-life events such as mode of birth, breastfeeding, and exposure to antibiotics and other factors such as air pollution, smoking, psychological state, exercise, and diet are among the potential environmental contributors of IBD development or disease activity [3]. 

Significant changes in dietary intake during the past decades have been associated with the increase in incidence of UC. The relationship between diet and UC development has been indicated in several epidemiological studies [4]. Two recent meta-analysis studies showed that soft drink consumption and sucrose intake were associated with 69% and 10% increased risk of UC development, respectively [5,6]. Consumption of fruits (odds ratio: 0.57) and vegetables (odds ratio: 0.71) was related to decreased odds of UC development in another meta-analysis study [7]. A significant association between meat intake (red meat in particular) and UC risk was found in a meta-analysis of seven epidemiological studies (summary relative risk: 1.47) [8]. Furthermore, whereas *n*-3 polyunsaturated fatty acids (PUFAs) content of diet was related to decreased odds of UC development (odds ratio: 0.56) [9], dietary arachidonic acid (an *n*-6 PUFA) as measured in adipose tissue increased risk of UC development (relative risk: 4.16) in a large prospective cohort study among Danish adults [10]. Although the exact pathophysiological mechanisms in which diet plays a role in IBD development remain unknown, several plausible explanations including its effects on composition of gut microbiota, production of microbial metabolites, alterations in mucosal immunity, and mucosal barrier function have been proposed [11] (Figure 1). 

Abnormalities in the intestinal microbiota have been reported in some, but not all UC patients [1,12,13,14]. In some studies, UC patients have been shown to have decreased bacterial diversity, characterized by a decreased Firmicutes and increased *Gammaproteobacteria* and *Enterobacteriaceae* [15]. However, it is not clear if bacterial dysbiosis is the cause or effect of mucosal inflammation in UC [1,13] (Figure 2). 

Dietary factors can be related to UC pathogenesis or disease course through direct effects on the host or indirect effects through modulations of composition or function of gut microbiota. Diet has a major role in shaping gut microbial composition [16]. For instance, increased *Bacteroidetes* and decreased *Firmicutes* and *Enterobacteriaceae* in rural African children in comparison to European children were mainly attributed to differences in dietary patterns between the two populations [17]. Therefore, it has been suggested that diet-induced changes in microbiota may transform healthy gut microbiota into a disease-inducing entity that could either initiate or perpetuate inflammation in patients with IBD [16]. Agus et al. [18] indicated that a high fat/high sugar diet resulted in intestinal mucosal dysbiosis characterized by an overgrowth of pro-inflammatory proteobacteria and a decrease in protective bacteria. In addition, they showed that the transplantation of feces from high fat/high sugar fed mice to germ-free mice increased susceptibility to adherent-invasive *Escherichia coli* infection. 

In addition to their significant effects on microbial composition, dietary factors can also affect the metabolic functions of gut microbiota. Short chain fatty acids (SCFA), which are defined as the groups of fatty acids with fewer than six carbons including formic acid (C1), acetic acid (C2), propionic acid (C3), butyric acid (C4), and valeric acid (C5), are derived from commensal bacterial fermentation of indigestible dietary fibers in both the small and large intestines [19,20]. Acetate, propionate and butyrate account for more than 95 % of all the SCFA content in the gut [19]. Acetate and butyrate in particular have an essential role in maintaining mucosal barrier function and modulating immune function [21,22] (Figure 3). SCFA regulate the functions of epithelial and/or immune cells through altering gene expression, cellular differentiation, chemotaxis, proliferation, and apoptosis [19]. The number of SCFA-producing bacteria such as *Faecalibacterium prausnitzii* is decreased in some UC patients, and these are inversely correlated with disease activity [23]. Moreover, a western diet characterized by high intake of sugar [18,24] and fat [18] and decreased amount of dietary fiber was associated with decreased SCFAs and increased susceptibility to colitis in experimental studies. 

The mucus layer and intestinal epithelium are the first physical and chemical barriers against intestinal bacteria, pathogens, and food antigens. A dysregulated mucosal immune response characterized by alterations in the innate immune system, activation of effector T-cells, increased presence of B-cells and antibody production, and increased production of pro-inflammatory mediators have a major role in the pathogenesis of IBD [1,25]. Dietary factors can have direct effects on host cells. For instance, it was shown that luminal iron may directly affect function of intestinal epithelial cells and T cells in addition to triggering epithelial cell stress-associated apoptosis [26]. Zinc is an important cofactor for various intestinal metalloproteinases, and zinc deficiency has been associated with reduced barrier integrity and increased permeability in IBD patients [27]. There is also increasing evidence for a role of vitamin D in strengthening the innate immune system and reducing inflammation in experimental and human IBD [28]. Relationships between PUFAs content of diet and inflammatory processes in IBD have also been shown [29,30]. Eicosapentaenoic acid and docosahexaenoic acid that are long chain dietary *n*-3 PUFAs inhibit genes that are involved in inflammatory process and alter the composition of cell membranes by displacing *n*-6 PUFAs, influencing lipid raft formation in cell signaling [30]. Dietary amino acids act as key regulatory factors in cellular and microbial metabolic pathways and play important roles in gut homeostasis. Intestinal inflammation as seen in IBD affects several metabolic pathways related to metabolism of amino acids [22]. It has been reported that several food additives, such as maltodextrin, emulsifying agents, or thickeners such as carboxymethyl cellulose, carrageenan, and xanthan gum, may also have detrimental effects on intestinal homeostasis as well [31]. A complete review of all dietary factors and host interactions is beyond the scope of this review; however, those interactions have been described comprehensively in other recent review articles [11,21,22,31,32,33,34]. 

Currently, dietary recommendations for management of IBD-related symptoms are scarce and non-evidence-based, mainly due to the limited number of dietary interventions in this population. In the present review article, we summarize findings from previously conducted dietary interventions in UC patients. 

## 2. Methods

An electronic search in MEDLINE (Ovid) from inception to April 1, 2019, was conducted in order to identify any dietary intervention studies on UC subjects). Reference lists of included studies were also checked to identify relevant studies that might have been missed during initial search in MEDLINE. Studies that only focused on nutritional supplements, enteral or parenteral nutrition, or were published in languages other than English were not included. A comprehensive full-text review of identified studies was conducted after the title screening and abstract screening of potentially relevant articles. Collected data included journal name, publication year, design of the study, age, sex, sample size, disease condition, intervention and comparator(s) of interest, outcome(s), outcome measures, and main findings. The MEDLINE search strategy was as follows:randomized controlled trial/clinical trial.pt.randomi?ed.ti,ab.placebo.ti,ab.randomly.ti,ab.trial.ti,ab.1 or 2 or 3 or 4 or 5 or 6Inflammatory Bowel Diseases/inflammatory bowel disease.tw.ibd.tw.ulcerative colitis.tw.colitis.tw.8 or 9 or 10 or 11 or 12Diet/diet*.tw.food.tw.14 or 15 or 167 and 13 and 17

## 3. Results

Our primary electronic search yielded 424 unique references. After title and abstract screening, nine studies were selected for full-text review. Following the full-text review, seven randomized controlled trials that met the inclusion and exclusion criteria were selected for this review [35,36,37,38,39,40,41]. The general characteristics of the included studies are summarized in Table 1. 

Wright et al. [35] randomly allocated UC patients with disease relapse into a milk-free diet (*n* = 26) (all milk and milk products, whether in the form of dairy products such as fresh milk and cheese or as powdered milk, were excluded, and butter was permitted), a gluten-free plus milk-free diet (*n* = 27), or a control group (*n* = 24). Patients were asked to follow the diets for one year after the induction of remission, and they were followed monthly to assess if they experienced disease relapse, which was defined as diarrhea with an average of four or more stools a day for at least a week and with macroscopic blood present, together with sigmoidoscopic evidence of inflammation. Although the relapse rate was higher in patients randomized to the control group in comparison to those on a milk-free diet (79.2% vs. 61.5%), it did not reach statistical significance (*p* = 0.2). In addition, the relapse rate in the gluten-free plus milk-free diet was 70.4%, which was comparable to that in the other two groups. 

In a small randomized controlled trial for 6 weeks [36], 18 adult UC patients with mild to moderate disease activity were randomized to a symptoms-guided elimination diet (*n* = 11) or a control group (*n* = 7). Patients in the control group were asked to document but not alter their dietary intake. However, patients in the experimental group were instructed to exclude foods that appeared to provoke their symptoms. Fried foods were prohibited. In addition, refined sugars, additives and preservatives, all condiments and spices other than salt, and beverages other than boiled water were prohibited during the 6-week trial for patients randomized to the elimination diet group. In the first week, dairy products were excluded from the diet, but were introduced over the next weeks in the following order: Skim milk, yogurt, skim-milk cheese, full-cream milk, cream, and full-cream cheese. Each week, subjects in the intervention group were interviewed in person, and their symptoms were reviewed in relation to the foods eaten during the previous week. The food menu was expanded over the 6-week trial to include as more variety of foods that each participant could tolerate. The induction of clinical remission rate (the passage of normal stools with absence of rectal bleeding) 6 weeks after the baseline visit was significantly higher in patients who received the symptoms-guided diet (36.3% vs. 0.0%). However, the endoscopic and histologic improvement was comparable between the two groups. 

In another study [37], children with newly diagnosed UC were randomly assigned to a cow’s milk protein (CMP) elimination diet (*n* = 14) or a normal diet as the control group (*n* = 15). The study aimed to compare the clinical remission rate between the two groups following the IBD induction therapy and the rate of clinical relapse (defined as the occurrence or worsening of symptoms accompanied by an increase of Pediatric Ulcerative Colitis Activity Index > 10 points which required treatment with corticosteroids, immunosuppressive agents, or surgery) between the two groups during the one-year trial. The authors reported that the clinical response rate four weeks after the initiation of the induction therapy was not different between the two groups (92.8% in CMP elimination diet vs. 80.0% in the control group, *p* = 0.6). In addition, clinical relapse rate was comparable between the two groups (53.8% in CMP elimination diet group vs. 53.3% in the control group). In addition, they found no significant changes in serum C-reactive protein (CRP), erythrocyte sedimentation rate or fecal calprotectin (FCP) in the two diet groups from baseline to the last visit. 

Kyaw et al. [38] recruited 112 adult UC patients and randomly assigned them to a dietary intervention and a control group. Patients in the intervention group were given an educational booklet that contained dietary recommendations to eat little and often (four to six times a day), drink adequate fluids, decrease excess intake of fat, decrease simple carbohydrates, and decrease high-fiber foods during flare. Patients were also advised to increase intake of “good-quality protein” during flare and eliminate dairy products if they were lactose intolerant. Patients randomized to the control group were provided with a booklet that included general recommendations on healthy eating (e.g., to choose higher fiber or whole grain carbohydrates, to eat lots of fruits and vegetables) and were assigned to follow their usual diet. At 24 weeks, there was a significant reduction in the Simple Clinical Colitis Activity Index (SCCAI) score in the intervention group compared with an increase in the score in the control group. However, there was no statistically significant change in quality of life scores from baseline to week 24 in the two groups. 

Bhattacharyya et al. [39] conducted a small, randomized, double-blind, placebo-controlled, multicenter, clinical trial on UC patients to investigate the effect of the common food additive carrageenan on clinical relapse rates. The authors recruited UC patients over the age of 18 in clinical remission (SCCAI ≤ 2). Patients randomized to the carrageenan group (*n* = 5) received the carrageenan-containing capsules (200 mg/day). Patients randomized to the placebo group received similar-appearing dextrose-containing capsules (*n* = 7). The study duration was 12 months, and participants were instructed to follow a carrageenan-free diet during that period. The primary outcome measure was occurrence of clinical relapse, which was defined as an increase of two (or more) points on the SCCAI in association with an increase in treatment. The Short Inflammatory Bowel Disease Questionnaire was used to assess changes in quality of life. In addition, blood and stool samples were collected to measure inflammatory markers. They found that UC patients who were on a carrageenan-free diet plus placebo had a lower relapse rate in comparison to patients who were on a similar diet plus two oral capsules of carrageenan per day (0.0% vs. 60.0%, *p* = 0.05). In addition, they reported that carrageenan consumption aggravated disease activity as indicated by increase in FCP (*p* = 0.06) and interleukin-6 (*p* = 0.02). However, there was no statistically significant difference between the two groups in terms of changes in quality of life scores. 

Pedersen et al. [40] conducted an open-label trial of patients with IBD (61 UC and 28 CD) in remission or with mild-to-moderate disease and coexisting IBS-like symptoms. Patients were randomly assigned to a low Fermentable, Oligosaccharides, Disaccharides, Monosaccharides and Polyols (FODMAP) diet (*n* = 44) or a normal diet (*n* = 45) for 6 weeks. In UC patients, there was a significant decrease in severity of IBS-related symptoms (assessed by IBS symptom Severity System) in both diet groups, and this response was not different between the two groups. However, the authors reported a significant decrease in disease activity assessed by SCCAI but only in patients randomized to the low-FODMAP diet. In addition, low-FODMAP diet increased quality of life of IBD patients (assessed by SIBDQ). However, low-FODMAP diet did not change CRP and FCP levels significantly. 

In a recent open-label, stratified study, Jian et al. [41] randomly allocated 97 UC patients who were in remission or had mild to moderate disease activity to a food exclusion group versus a sham diet group. At baseline, the presence of blood IgG antibodies specific to egg, wheat, milk, corn, tomato, crab, rice, soybean, cod, shrimp, mushrooms, beef, chicken, and pork antigens were tested. Based on IgG antibody titers, patients randomized to the exclusion diet group were instructed to stop or reduce taking specific food items. Patients in the control group were asked to follow their routine diet. The duration of the trial was 6 months. They reported that in comparison to the control diet, following the exclusion diet was associated with a significant decrease in Mayo scores and improvement in quality of life. 

## 4. Discussion

It has been suggested that environmental factors including diet play an important role in the pathophysiology of IBD and especially in UC, a chronic colonic inflammation. In the present article, after a brief overview of potential mechanisms in which diet plays a role in the pathogenesis of IBD, we then reviewed dietary intervention studies in UC patients.

The three randomized controlled trials that have been performed to assess the efficacy of dietary interventions for maintenance of remission in UC [35,37,39] were all focused on complete exclusion of one or more food items that were hypothesized to trigger IBD symptoms. Two of these studies [35,37] aimed to eliminate milk or dairy products; however, they failed to show a significant decrease in relapse rate in patients randomized to the elimination diet group in comparison to those randomized to the control diet. This finding is important as unnecessary dietary restrictions that lack supporting scientific evidence may result in several nutritional deficiencies (e.g., calcium due to exclusion of milk and dairy products) in IBD patients [42]. Therefore, patients should be informed by their health care team about the possible harmful effects of food elimination diets. 

In the present review, the only elimination diet that was associated with a reduction in clinical relapse rate in UC patients who were in remission at baseline was a carrageenan-free diet [39]. Carrageenan belongs to a family of sulfated polysaccharides and are extracted from seaweeds. It is approved as “generally recognized as safe” by the United States Food and Drug Administration and is used in the food industry for its gelling, thickening, and stabilizing properties. It has been suggested that carrageenan may reduce protein and peptide bioaccessibility, disrupt normal epithelial function, and promote intestinal inflammation [43]. However, others have been skeptical about these findings, which are mainly derived from experimental animal studies [44]. The results from the randomized clinical trial in which a carrageenan-free diet was found to be related to lower relapse rate and decreased inflammation (as assessed by decreased serum interleukin-6 and FCP) should be interpreted with caution as the sample size of this multi-center trial was very small (*n* = 12) and the reported *p*-values obtained from parametric tests were marginally significant. Therefore, these interesting findings need to be confirmed in future well-powered randomized controlled studies. 

We identified only two studies [36,37] that tested the efficacy of diet for induction of remission in UC patients. In the first study, exclusion of food items that were found to trigger UC-related symptoms was associated with higher clinical remission rate in comparison to a normal diet [36]. Although the elimination of foods was based on each participant’s self-reported food intolerance, there were some general recommendations regarding specific food groups/items such as dairy products, refined sugar, and beverages. However, the study was performed on a small number of patients (*n* = 18), and the duration of follow-up was short (6 weeks). In addition, the intervention did not result in endoscopic or histologic improvement in that time period. Furthermore, patients in the intervention group experienced a mean weight loss of 2.5 kg that was not explained in the study. The authors also reported that there was no food that triggered symptoms in all patients. However, spicy and curried foods and fruits (specially grapes, melon, and citruses) were commonly reported to provoke symptoms. In the second study, which was performed in pediatric UC patients with active disease, elimination of cow milk protein from diet was not beneficial neither for induction or for maintenance of remission during a one-year follow-up in comparison to a control diet [37]. As mentioned by the authors, the dietary restrictions that many IBD patients follow often are not supported by scientific evidence. These inappropriate diets reduce caloric intake and may contribute to malnutrition and micronutrient deficiencies, especially in pediatric patients. Whether a subgroup of patients with UC (e.g., patients with lactose intolerance or atopy) will benefit from elimination diets or not needs to be explored in future clinical trials.

In this review, we also included three other studies that recruited patients with active disease and UC remission concurrently. They reported the effectiveness of comprehensive dietary advices [38], low FODMAP [40] or IgG-guided exclusion diets [41] in reduction of disease activity in UC patients. Although these findings are encouraging, one of the major limitations of these studies is that they did not report their findings for patients with active disease versus patients in UC remission separately to allow meaningful interpretations [45]. Therefore, we suggest that in future studies the dietary interventions be focused on clearly specified groups of patients (e.g., active disease or in remission) or study outcomes to be reported for different groups of participants separately.

Diet is of major interest for IBD patients, and they use a variety of dietary strategies to manage their underlying disease and related symptoms [34]. Despite the significant role of diet in the development of IBD or management of gastrointestinal symptoms in these patients, we could identify only a few randomized controlled trials that assessed the efficacy of diet for induction of remission, maintenance of remission, or improvement of gastrointestinal symptoms in UC patients. In addition, in the previous studies, the underlying mechanisms in which diet may prevent increases in colonic or systemic inflammation and ultimately help patients to maintain remission have not been investigated. There are many omics fields involved in the study of pathogenesis of IBD such as genomics, metagenomics, transcriptomics, proteomics, and metabolomics [32]. As dietary factors have a significant impact on some of these key players of IBD development, investigating the changes in this multi-omic network of IBD during a controlled dietary intervention has the potential to elucidate the underlying mechanisms of diet-IBD interactions. High quality, well-powered human dietary intervention studies for management of IBD may include the following: Quantification of baseline habitual diet using appropriate tools such as food frequency questionnaires, monitoring of adherence to the diet using food recalls/records, large long-term controlled trials, use of a control diet to determine the specificity of observed effects to the intervention, use of a variety of subjective and objective endpoints (e.g., symptoms, quality of life, clinical biomarkers, endoscopic and histological evaluations) to monitor response to dietary interventions [34], and consider the use of omic-based assessments of serum, urine, stool, and/or intestinal biopsies to investigate underlying protective mechanisms. Considering findings from previous observational studies and clinical trials, investigating the potential benefits of following a healthy dietary pattern, such as experimental anti-inflammatory diets that incorporate several dietary recommendations, is of great value in the management of UC-related symptoms and inflammation. Furthermore, as indicated in elimination diet studies, food intolerances are individual-based, and not all patients will benefit from excluding certain food items/groups. Therefore, personalized dietary recommendations that take into account each patient’s food intolerances and food preferences should be the subject of future well-designed dietary trials in IBD patients.

## 5. Conclusions

In conclusion, we found that there have been few well-designed and/or adequately powered randomized clinical trials to investigate the role of diet in maintenance of remission in UC patients. As suggested in a recent Cochrane systematic review [45], consensus on the composition of evidence-based dietary interventions in IBD patients is required and there is a need for more high-quality, well-powered, randomized, controlled trials to assess the efficacy of these interventions. 

## Figures and Tables

**Figure 1 nutrients-11-01498-f001:**
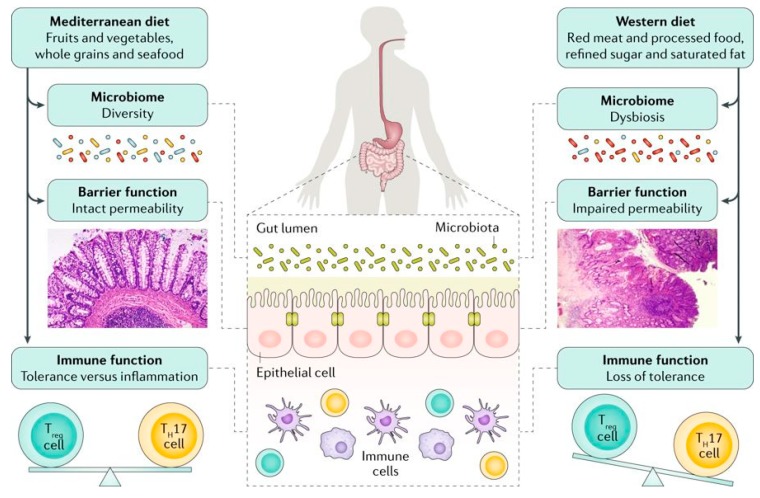
Although the exact mechanisms responsible for the association between diet and development of inflammatory bowel disease is unknown, several mechanisms have been suggested. An unhealthy dietary pattern such as a Western diet has been linked to changes in the gut microbiome and epithelial barrier function and seems to have a direct influence on immune function, triggering a pro-inflammatory environment characterized by an imbalance in the T helper 17 (T_H_17) cell to regulatory T (T_reg_) cell ratio [Adapted with permission [11]].

**Figure 2 nutrients-11-01498-f002:**
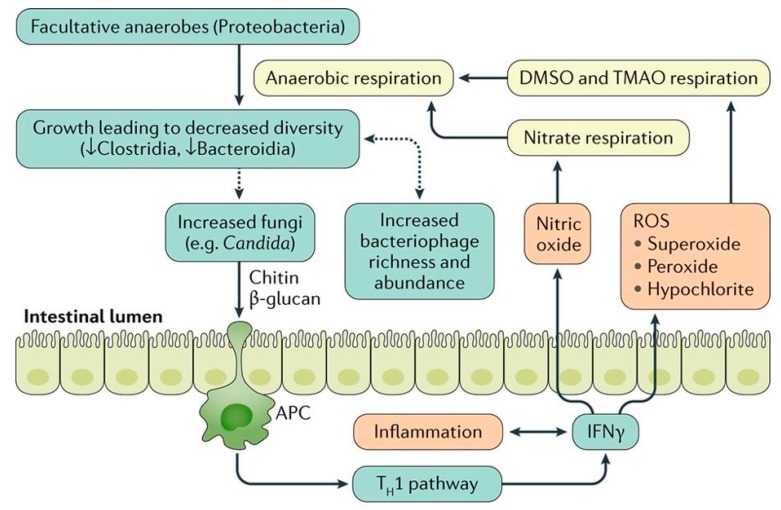
The relationship between gut microbiota and colonic inflammation in inflammatory bowel disease. Inflammation in colon stimulates production of Interferon gamma (IFN-γ) that eventually generates reactive oxygen species (ROS). ROS make products for anaerobic respiration. These products can be used by facultative anaerobes to outgrow, which leads to decreased bacterial diversity. The dysbiotic microbiota may further stimulate the growth of fungi that can worsen inflammation via chitin and β -glucan antigen-presenting cell (APC) activation of the type 1 T helper (T_H_1) pathway. In addition, the microbial dysbiosis is associated with increased bacteriophage richness and abundance, which can affect the bacterial microbiota via gene transfer. DMSO, dimethyl sulfoxide; TMAO, trimethylamine N-oxide. [Adapted with permission [13]].

**Figure 3 nutrients-11-01498-f003:**
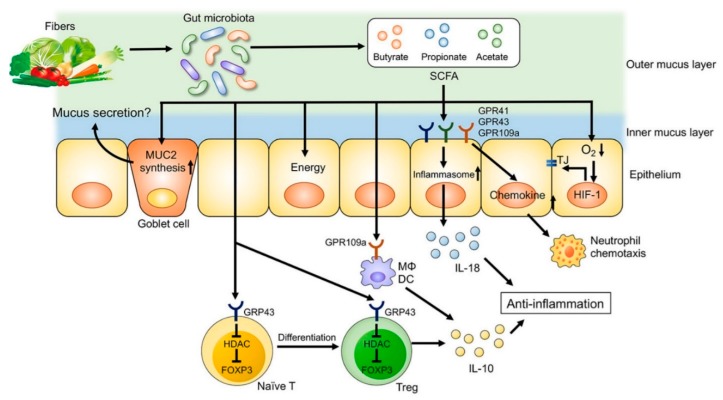
The role of fiber-derived short chain fatty acids (SCFAs) in regulation of intestinal homeostasis. SCFAs serve as energy substrates for colonocytes. In addition, SCFAs regulate intestinal barrier function and immune system through G-protein-coupled receptors (GPRs) signaling. SCFAs promote the differentiation of regulatory T (Treg) cells and the production of interleukin (IL)-10 through GPR43. Furthermore, SCFA facilitate inflammasome activation in colonic epithelial cells through GPR43, stimulating IL-18 production that is critical for anti-inflammation and epithelial repair. SCFAs also regulate intestinal barrier function via enhancing the expression of tight junction proteins and the synthesis of mucin (MUC)2. DC, dendritic cells; FOXP3, forkhead box P3; HDAC, histone deacetylases; Mϕ, macrophages; TJ, tight junctions. [Adapted with permission [22]].

**Table 1 nutrients-11-01498-t001:** General characteristics of studies examining the role of diet for maintenance of remission in ulcerative colitis patients.

First Author (Year)	Country	Study Design	Population	Intervention/Comparator(s) (Sample Size ^1^)	Duration	Outcomes and Assessment Tools
Wright (1965) [35]	UK	Randomized controlled clinical trial	Adult UC patients in clinical remission after induction of remission	Milk-free diet (*n* = 26)/gluten-free plus milk-free diet (*n* = 27)/“dummy diet” as control (*n* = 24)	12 months	Relapse: Symptoms + sigmoidoscopy, biopsy, dietary adherence: interview
Candy (1995) [36]	South Africa	Randomized, controlled clinical trial	Adult UC patients with mild to moderate disease activity	Symptoms-guided elimination diet (*n* = 11)/normal diet as control (*n* = 7)	6 weeks	Induction of clinical remission, sigmoidoscopy, histopathology, dietary adherence: interview
Strisciuglio (2013) [37]	Italy	Single-center, randomized, controlled clinical trial	Pediatric newly diagnosed UC patients	Cow's milk protein elimination diet (*n* = 14)/normal diet as control (*n* = 15)	12 months	Induction of clinical remission, clinical relapse: PUCAI, Physician global assessment, serum C-reactive protein, erythrocyte sedimentation rate, fecal calprotectin, endoscopic evaluation, histological evaluation, dietary adherence: food diaries
Kyaw (2014) [38]	UK	Randomized, controlled clinical trial	Adult UC patients	Comprehensive dietary advices (*n* = 61)/general dietary recommendations +normal diet as control(*n* = 51)	24 weeks	Disease activity: SCCAI, quality of life: IBDQ, dietary adherence: food frequency questionnaire
Bhattacharyya (2017) [39]	USA	Randomized, double-blind, placebo-controlled, multicenter, clinical trial	Adult UC patients in clinical remission	No-carrageenan diet + carrageenan-containing capsules (200 mg/d) (*n* = 7)/no-carrageenan diet + placebo (dextrose) (*n* = 7)	12 months	Clinical relapse: SCCAI, quality of life: SIBDQ, serum cytokines, fecal calprotectin, dietary adherence: 24 h dietary recalls
Pedersen (2017) [40]	Denmark	Randomized, open-label, controlled clinical trial	Adult UC patients in remission, or mild to moderate disease activity and coexisting IBS-like symptoms	Low FODMAP diet (*n* = 44)/normal habitual diet as control (*n* = 45)	6 weeks	Disease activity: SCCAI, Severity of IBS symptoms: IBS-SSS, quality of life: SIBDQ, C-reactive protein, fecal calprotectin, dietary adherence: food frequency questionnaire
Jian (2018) [41]	China	Randomized, open-label, stratified clinical trial	Adult UC patients in remission, or mild to moderate disease activity	Immunoglobulin G-guided exclusion diet (*n* = 49)/normal diet as control (*n* = 48)	6 months	Disease activity: Mayo score, quality of life: IBDQ, body mass index, albumin, transferrin, prealbumin, extraintestinal manifestation of the disease, food-specific IgG antibodies, dietary adherence: food diaries

UC: ulcerative colitis; PUCAI: Pediatric Ulcerative Colitis Activity Index; SCCAI: Simple Clinical Colitis Activity Index; SIBDQ: Short Inflammatory Bowel Disease Questionnaire; IBS: Irritable bowel syndrome; FODMAP: Fermentable, Oligosaccharides, Disaccharides, Monosaccharides and Polyols; IBS-SSS: IBS symptom severity system; IBDQ: Inflammatory Bowel Disease Questionnaire. ^1^ Number of patients used for statistical analysis.

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
