# Peer review of "Diet in the Pathogenesis and Management of Ulcerative Colitis; A Review of Randomized Controlled Dietary Interventions"

_nutrients, 2019, doi:10.3390/nu11071498_

Round 1

Reviewer 1 Report

This paper summarizes the seven randomized controlled dietary intervention studies in ulcerative colitis (UC) that they can find in the litterature. The overall conclusion is that although these papers contain a number of interesting observations they do not provide a basis for generalized dietary recommendations in UC. Nethertheless it is worthwhile to summarize the studies, which differ considerably in design and focus, as a basis for further studies.

Criticisms: 1. In the introduction the authors list a number of external factors that may influence risk and course of IBD. It is stated that these factors together with diet are "..among the potential environmental contributors of IBD  development or disease activity (ref 6). Among all factors the protective effect of smoking in UC, the negative effects of smoking in Crohn´s disease, the negative effects of NSAID, and the acute onset of UC after bacterial gut infections are more well established factors than the other ones. Could the authors be more specific about the epidemioölogical evidence for the dietary influence, when they refer to ref 7? Values for relative risk? Generalizability of the findings in different studies? Figure 1 is fine, but the evdience behind it could be briefly summarized in a better way.

2. Must every article that deals with common and important diseases introduce the disease by telling how many billion dollars the healthcare costs for the disease are? This is not a health economy paper.

3. The bacteria, the mucous layer and the immune dysregulation are emphasized. The authors also mentioned other potential mechanisms by which diet might influence UC risk and course. What about a dietary influence on Paneth cell bactericidal enzyme secretion, on fluid and electrolyte secretion and on expression of mucosal brush border enzymes as alkaline phosphatase that dephosphorylates endotoxin?

4. The authors nicely discuss how future studies in this area need to be designed. The shortage of well controlled dietary studies in UC has, however, several causes. The UC patients are interested in their diet but also have a lot of subjective experiences and opinions. Todays medical treatment is rather effective. A rather large number of patients and a rather long follow up period is thus necessary to test the hypothesis that a specific diet decreases number of relapses. Could the authors go deeper into their study priorities. Which diet should first be compared with the generally healthy Mediterranian style diet? Each comparison will require large resources.   

Author Response

Comment: In the introduction the authors list a number of external factors that may influence risk and course of IBD. It is stated that these factors together with diet are "..among the potential environmental contributors of IBD  development or disease activity (ref 6). Among all factors the protective effect of smoking in UC, the negative effects of smoking in Crohn´s disease, the negative effects of NSAID, and the acute onset of UC after bacterial gut infections are more well established factors than the other ones. Could the authors be more specific about the epidemioölogical evidence,e for the dietary influence, when they refer to ref 7? Values for relative risk? Generalizability of the findings in different studies? Figure 1 is fine, but the evdience behind it could be briefly summarized in a better way.

Answer: In the revised manuscript we have provided findings from recent meta-analysis studies of epidemiological studies on the role of major dietary factors in the development of ulcerative colitis. We have also included values for relative risks or odds ratios.

 Comment: Must every article that deals with common and important diseases introduce the disease by telling how many billion dollars the healthcare costs for the disease are? This is not a health economy paper.

Answer: We have deleted the parts about the healthcare costs of inflammatory bowel disease as you suggested.

Comment: The bacteria, the mucous layer and the immune dysregulation are emphasized. The authors also mentioned other potential mechanisms by which diet might influence UC risk and course. What about a dietary influence on Paneth cell bactericidal enzyme secretion, on fluid and electrolyte secretion and on expression of mucosal brush border enzymes as alkaline phosphatase that dephosphorylates endotoxin?

Answer: As you know the pathophysiological mechanism in which dietary factors are related to the pathogenesis of IBD are very complex. The main aim of this review was to summarize findings from previously conducted randomized controlled dietary intervention for management of symptoms in UC patients. Therefore, for the pathophysiological mechanisms of dietary determinants of UC development we only focused on major factors for this disease and cited other recently published review articles that focused mainly on the underlying mechanisms. Paneth cell dysfunction is more related to Crohn’s disease. In addition, describing all the mechanisms would make the manuscript much longer (the second reviewer had an issue with the length of the introduction section). 

Comment: The authors nicely discuss how future studies in this area need to be designed. The shortage of well controlled dietary studies in UC has, however, several causes. The UC patients are interested in their diet but also have a lot of subjective experiences and opinions. Todays medical treatment is rather effective. A rather large number of patients and a rather long follow up period is thus necessary to test the hypothesis that a specific diet decreases number of relapses. Could the authors go deeper into their study priorities. Which diet should first be compared with the generally healthy Mediterranian style diet? Each comparison will require large resources.  

Answer: Thanks for your comment. So far the usefulness of following a healthy dietary pattern such as a Mediterranean diet in the management of UC/IBD has not been investigated in the context of IBD. Therefore, it would be valuable to investigate their potentials in controlling symptoms and inflammation in IBD patients in future study as a potential intervention (not as a control diet). 

Reviewer 2 Report

This is a very interesting article! This field and probably it will be more explored in the future also in the context of gut microbiota (both in feces and mucosa). I think the article would benefit of an information that exclusive enteral nutrition is used as a first linte treatment as induction of remission in pediatric Crohns disease; also Thank ypu for an interesting in kids with colonic CD. This is important in the context to the study of Cleyen et al. Lancet 2016. (Inherited determ.of CD and UC phenotypes..) Now it is difficult to follow WHY this angle (IBD/UC treated with diet) should be further explored.

ABSTRACT: Sentence 1: I suggest a more moderate statement; "one of the suggested environmental contributors of IBD, in particular studied in Crohn's disease"

INTRODUCTION: I think the introduction is too long. In order to keep it shorter sentence 4 and 5 can be skipped.

Figure 2 and 3 are very interesting; although I think the article would be more lucid without them; ´they are complicated and the mechanisms they show is already described in the text. The introduction is too long, it is necessarily not important to describe the mechanisms in detail.

The last sentence in the introduction is not coherent with the first in the abstract (that's why is suggest the re-writing of the first abstract sentence.

METHODS:

In the Wright study; it is necessary that the authors describe what "low-roughage diet" means.

In the next study presented (Candy, 1995) the authors need to describe what " symptoms-guided elimination diet" means and what the control-group ate. Now the reader can't follow what the RCT descriminate between.

In the nr 4 study (Kyaw et al) a little more information of what the control groups information about healthy eating means.

In Bhattacharyya et al, something is missing in the sentence started with "The Short IBD Questionnaire…"

DISCUSSION: The first sentence is veeeery long. Take the second sentence and make it the first.

I think it is hard to follow the sentence " Three RCT have been performed to assess… Is not the FODMAP study, with impact on IBD patients; not a dietary intervention? Or is this study not an RCT (described as an open-label trial).

I would like the section about Candy et al, would be a more described. This review is about UC and dietary intervention and therefore, the two studies that describes diet for induction of remission, could be more discussed.

Author Response

Comment: This is a very interesting article! This field and probably it will be more explored in the future also in the context of gut microbiota (both in feces and mucosa). I think the article would benefit of an information that exclusive enteral nutrition is used as a first linte treatment as induction of remission in pediatric Crohns disease; also Thank ypu for an interesting in kids with colonic CD. This is important in the context to the study of Cleyen et al. Lancet 2016. (Inherited determ.of CD and UC phenotypes..) Now it is difficult to follow WHY this angle (IBD/UC treated with diet) should be further explored.

Answer: Thanks for showing interest to our manuscript. Since the focus of this manuscript was on dietary intervention studies in adults with UC, we decided to exclude those studies that focused on exclusive enteral nutrition, such as published in pediatric Crohn’s disease. This approach is similar to other review articles including a recently published Cochrane systematic review (Reference # 42) on this subject. In addition, the reference to the article regarding genetics in IBD by Cleyen et al, as suggested by the reviewer, was not included, as the focus of our manuscript was mainly on environmental factors in IBD, esp. in UC.

Comment: ABSTRACT: Sentence 1: I suggest a more moderate statement; "one of the suggested environmental contributors of IBD, in particular studied in Crohn's disease"

Answer: We have edited the section accordingly.

Comment: I think the introduction is too long. In order to keep it shorter sentence 4 and 5 can be skipped. Figure 2 and 3 are very interesting; although I think the article would be more lucid without them; ´they are complicated and the mechanisms they show is already described in the text. The introduction is too long, it is necessarily not important to describe the mechanisms in detail.

Answer: We have deleted a few sentences from the introduction section (healthcare costs of IBD). Although the main aim of this manuscript was to summarize the previously published dietary intervention on UC patients, we aimed to briefly explain the pathophysiological mechanisms in which diet plays a role in the development of IBD. Furthermore, we found it difficult to shorten this section as two other reviewers suggested to keep the figures and even explain some parts in the introduction section in more detail!

Comment: The last sentence in the introduction is not coherent with the first in the abstract (that's why is suggest the re-writing of the first abstract sentence

Answer: We have revised the first sentence of the abstract as you suggested.

Comment: In the Wright study; it is necessary that the authors describe what "low-roughage diet" means.

Answer: Unfortunately, there was no explanation about the term “low-roughage” in their manuscript. We have deleted this term in the revised manuscript but have provided more details about the diet as you suggested.

Comment: In the next study presented (Candy, 1995) the authors need to describe what " symptoms-guided elimination diet" means and what the control-group ate. Now the reader can't follow what the RCT descriminate between.

Answer: Thanks for your comment. More details have been provided in the revised manuscript.

Comment: In the nr 4 study (Kyaw et al) a little more information of what the control groups information about healthy eating means.

Answer: This information was added.

Comment: In Bhattacharyya et al, something is missing in the sentence started with "The Short IBD Questionnaire…"

Answer: We have edited the sentence as you suggested. 

Comment: DISCUSSION: The first sentence is veeeery long. Take the second sentence and make it the first.

Answer: We have revised this part according to your suggestion. 

Comment: I think it is hard to follow the sentence " Three RCT have been performed to assess… Is not the FODMAP study, with impact on IBD patients; not a dietary intervention? Or is this study not an RCT (described as an open-label trial).

Answer:  Those three studies were conducted specifically for “maintenance of remission”. However, the FODMAP study which included patients with active and inactive disease was performed for management of symptoms and inflammation in general.

Comment: I would like the section about Candy et al, would be a more described. This review is about UC and dietary intervention and therefore, the two studies that describes diet for induction of remission, could be more discussed.

Answer: Thanks for your comment. These two studies have been described in more detail in the revised manuscript.

Reviewer 3 Report

I found the manuscript interesting considering the fact that it revises the randomized controlled dietary interventions carried out on patients affected by ulcerative colitis. As well as mentioned in other reviews concerning IBDs, in this review the urgent need of further studies in this field emerges quite clearly. I think that the information provided in the manuscript is sufficient, despite I would expect more bibliographic research, and the figures are well organized and represented.

Author Response

Comment: I found the manuscript interesting considering the fact that it revises the randomized controlled dietary interventions carried out on patients affected by ulcerative colitis. As well as mentioned in other reviews concerning IBDs, in this review the urgent need of further studies in this field emerges quite clearly. I think that the information provided in the manuscript is sufficient, despite I would expect more bibliographic research, and the figures are well organized and represented.

Answer: Thanks a lot for showing interest to our manuscript. We have optimized the bibliographic research in this manuscript.